# Integrated DNA Copy Number and Expression Profiling Identifies IGF1R as a Prognostic Biomarker in Pediatric Osteosarcoma

**DOI:** 10.3390/ijms23148036

**Published:** 2022-07-21

**Authors:** Aaron M. Taylor, Jiayi M. Sun, Alexander Yu, Horatiu Voicu, Jianhe Shen, Donald A. Barkauskas, Timothy J. Triche, Julie M. Gastier-Foster, Tsz-Kwong Man, Ching C. Lau

**Affiliations:** 1The Jackson Laboratory for Genomic Medicine, Farmington, CT 06032, USA; aaron.taylor@jax.org; 2Department of Pediatrics-Oncology, Baylor College of Medicine, Houston, TX 77030, USA; monikasun88@gmail.com (J.M.S.); alexanderyu@me.com (A.Y.); jjshen77024@gmail.com (J.S.); tman@bcm.edu (T.-K.M.); 3Program of Quantitative & Computational Biosciences, Baylor College of Medicine, Houston, TX 77030, USA; 4Dan L. Duncan Cancer Center-Bioinformatics, Baylor College of Medicine, Houston, TX 77030, USA; voicu@bcm.edu; 5Department of Population and Public Health Sciences, Keck School of Medicine, University of Southern California, Los Angeles, CA 90033, USA; barkausk@usc.edu; 6Pathology and Laboratory Medicine, Keck School of Medicine, University of Southern California, Los Angeles, CA 90033, USA; triche@usc.edu; 7Cancer and Hematology Center, Texas Children’s Hospital, Houston, TX 77030, USA; jxgastie@texaschildrens.org

**Keywords:** osteosarcoma, bone, pediatric, cancer, treatment, biomarker, metastasis, prognosis

## Abstract

Osteosarcoma is a primary malignant bone tumor arising from bone-forming mesenchymal cells in children and adolescents. Despite efforts to understand the biology of the disease and identify novel therapeutics, the survival of osteosarcoma patients remains dismal. We have concurrently profiled the copy number and gene expression of 226 osteosarcoma samples as part of the Strategic Partnering to Evaluate Cancer Signatures (SPECS) initiative. Our results demonstrate the heterogeneous landscape of osteosarcoma in younger populations by showing the presence of genome-wide copy number abnormalities occurring both recurrently among samples and in a high frequency. Insulin growth factor receptor 1 (IGF1R) is a receptor tyrosine kinase which binds IGF1 and IGF2 to activate downstream pathways involved in cell apoptosis and proliferation. We identify prevalent amplification of IGF1R corresponding with increased gene expression in patients with poor survival outcomes. Our results substantiate previously tenuously associated copy number abnormalities identified in smaller datasets (13q34+, 20p13+, 4q35-, 20q13.33-), and indicate the significance of high fibroblast growth factor receptor 2 (FGFR2) expression in distinguishing patients with poor prognosis. FGFR2 is involved in cellular proliferation processes such as division, growth and angiogenesis. In summary, our findings demonstrate the prognostic significance of several genes associated with osteosarcoma pathogenesis.

## 1. Introduction

Pediatric primary bone cancer is a rare malignancy that accounts for 6% of all cancers from birth to the age of 20 [1]. Osteosarcoma, a mesenchymal malignant tumor with an incidence rate of approximately 400 cases a year in the United States [1], is the most common bone cancer in this group. Most osteosarcoma cases are sporadic, with mutations in TP53 and RB1 common to 40% or more of these tumors [2]. Chromosomal defects due to chromosomal instability are very common in osteosarcoma, but no translocations are particularly disease-defining [3].

There has been significant previous research into the molecular characteristics of osteosarcoma, which has focused most heavily on the Notch and Wnt pathways. It has been shown that inhibiting Wnt signaling is effective at limiting both tumorigenesis and the metastatic potential of osteosarcoma in a murine model [4,5], and that the inhibition of Notch signaling decreases cell proliferation and tumor size in cells and mice, respectively [6]. A recent study which integrated copy number, methylation, and gene expression profiling on 10 primary tumor samples identified RUNX2, DOCK5, TNFSRF10A/D, and several Histone cluster 2 genes with possible roles in osteosarcoma tumorigenesis [7]. The overexpression of ezrin, a cytoskeletal linker protein, has a vital role in the metastatic potential of osteosarcoma and was associated with poor prognosis in pediatric patients [8]. Integrated analyses of gene expression, copy number, and methylation profiles in osteosarcoma cell lines has revealed a potential role for gene silencing and chromosomal aberrations in osteosarcoma tumorigenesis, particularly in the known oncoprotein c-Myc [7,9].

Micro-RNAs (miRNAs) are emerging as a critical factor in the formation and proliferation of osteosarcoma, with members of the miRNA-29 family such as miR-29a, miR-29b, and miR-29c playing an essential role [10]. miR-210 has also been found to be upregulated in pediatric osteosarcoma with prognostic implications [11]. Further work has shown that micro-RNA (miRNA) expression levels can be used to predict response to therapy [12,13], particularly the maternally expressed miRNA in the 14q region [14]. miRNAs involved in the Notch and Wnt pathways are also frequently aberrantly expressed, with Notch regulator miR-199b-5p playing a direct role in osteosarcoma pathogenesis [15,16]. Several other miRNAs are being explored due to their known dysregulation in osteosarcoma for potential therapeutic and diagnostic applications [17].

Despite the above advances in the molecular mechanisms involved in pediatric osteosarcoma, an overall survival (OS) rate of 68% has remained relatively unchanged over the past several decades [18]. Current treatment for pediatric osteosarcoma consists mainly of surgical tumor removal and chemotherapy. This treatment has been optimized in recent years to include a common regimen of neoadjuvant chemotherapy, surgery, radiation, and post-operative chemotherapy [2]. Neoadjuvant chemotherapy consists of a non-specific three-drug regime of high-dose methotrexate, doxorubicin, and cisplatin [2]. Despite the common application of neoadjuvant chemotherapy in treatment centers in the USA and abroad, it has not been shown to have any significant effect on event-free survival (EFS) [19], although tumor necrosis following therapy correlates with EFS [20]. The presence of metastases at diagnosis, particularly common in the lung, remains the most effective prognostic indicator for patient survival [21]. Clinical trials to determine the effectiveness of potential targeted chemotherapeutic agents based on results from previous studies have so far been unsuccessful [22,23], and have failed to discover a more effective treatment regimen for high-risk patients.

In order to discover prognostic biomarkers with potential therapeutic applications, we collected samples from 226 unique patients and performed array-based copy number and gene expression profiling. While previous studies suffered from a limited sample size, a lack of clinical information, or a reliance on cell lines, this study uniquely utilizes a large cohort of primary tumor samples with paired patient survival data. We identified biomarkers in both copy number and expression platforms associated with survival, and integrated these data types in order to determine how expression is altered by copy number changes. Using these methods, we determined that IGFR1 is frequently overexpressed due to large increases in copy number changes, and that both copy number and expression can be used as prognostic biomarkers and potential therapeutic targets.

## 2. Results

### 2.1. NMF Clustering of Gene Expression Data Identifies Two Distinct Clusters

Non-Negative Matrix Factorization (NMF) analysis was performed on the 103 samples profiled using the Human Exon 1.0 ST (HuEx) array and we identified two distinct clusters (Appendix A). To understand the biology underlying the clusters, differentially expressed genes which contributed to the NMF features were extracted (Figure 1, Appendix A). The top 10 processes overrepresented in the differentially expressed gene list were derived using MetaCore Pathway Analysis (Table 1). All features present on the Affymetrix Human Exon 1.0 ST array were used as the background. Differentially expressed genes involved in cell adhesion (including integrin priming, leucocyte chemotaxis, and cadherins) and development (regulation of angiogenesis, blood vessel morphogenesis, ossification and bone remodeling, regulation of epithelial-to-mesenchymal transition, and skeletal muscle development) pathways were overexpressed. Although the NMF clusters were significantly associated with both event-free (Wald test *p* = 0.0085) and overall (Wald test *p* = 0.0438) survival, adding initial metastatic status to the model abrogated their prognostic ability (Appendix A), suggesting that the NMF clusters may be a surrogate for initial metastatic status (Figure 1).

### 2.2. mRNA Expression Predicts Clinical Outcomes

In order to determine which genes were associated with clinical outcomes that are independent of initial metastasis, which is an existing prognostic factor at diagnosis, the sample set profiled on the Human Exon 1.0 ST array was utilized as a discovery cohort (103 samples), and a separate cohort profiled using the U133 Plus 2.0 array was used for validation (64 samples). The top 2.5% (437) most variable features (Appendix A) were utilized for survival analysis. For each feature, multivariate Cox proportional hazards models for both event-free survival (EFS) and overall survival (OS) time were created. A total of seven features were significantly associated after false discovery correction with overall survival (Benjamini–Hochberg-corrected Wald test, *p* < 0.05). Using only these features for the analysis, they were also significantly associated with event-free survival before false discovery correction across all genes. High expression of these features was associated with better prognosis (Appendix A).

To validate our results, probesets from the U133 array mapping to the same genes as the prognostically significant HuEx features were selected for survival analysis. A total of four U133 features, mapping to the same genes as three HuEx features, showed prognostic significance for both overall and event-free survival (Table 2). These features mapped to fibroblast growth factor receptor 2 (FGFR2), fin bud initiation factor homolog (FIBIN), pleiotrophin (PTN), and diacylglycerol kinase iota (DGKI). Higher FGFR2 and FIBIN expression is associated with better survival outcomes in both the discovery and validation set. The HuEx feature that matched to both PTN and DGKI correlated with better prognosis in the discovery set, as was the DGKI feature in the validation set. However, PTN expression in the validation set was deleterious. Samples with high expression of FGFR2 showed significantly better prognosis (Figure 2).

### 2.3. Characterization of Osteosarcoma Copy Number Aberrations

To identify DNA copy number aberrations associated with osteosarcoma, segmentation data from 147 osteosarcoma DNA samples were analyzed using GISTIC (Appendix A, Appendix A). In summary, we observed 77 significantly altered copy number regions composed of 25 amplifications and 52 deletions (residual q-value < 0.05). As a comparison, we contrasted our results with several osteosarcoma array CGH and probe-based studies [24,25,26,27]. Of the top 10 recurrent amplifications or deletions, 7 and 2, respectively, were also reported in previous studies (Table 3). Among these regions, the minimum proportion of samples with at least a moderate level of amplification or deletion of that copy number aberration was 33.3%, an indication of a highly rearranged genome that is characteristic of pediatric osteosarcoma patients. Recurrent copy number changes were identified in regions containing canonical osteosarcoma-associated genes, such as MYC and RB1, as well as in more novel regions such as 13q34.

A majority of the top 10 most frequently highly amplified regions have been reported, in previous studies including 17p11.2 (TOP3A, FLI1), 8q24.21 (MYC), 1q21.3 (MCL1, BNIPL), 19p13.2 (JUNB, NFIX), 6p21.1 (CDC5L), 19q12 (CCNE1) and 15q26.3 (IGF1R). High-frequency amplification of 17p11.2 was observed in 30.6% (45/147) of cases. This region encompasses topoisomerase III alpha (TOP3A), a key gene involved in many DNA structural maintenance processes and the transcription factor friend leukemia virus integration 1 (FLI1). Using segmentation summarized copy number values, we found that TOP3A and FLII are very positively correlated (rho > 0.5 for comparisons in both genes) with gene expression in both HuEx and U133 platforms. Recurrent copy number gain of 8q24.21 involving the transcription factor coding gene MYC is associated with higher expression in both datasets. Out of all the cases, 17% (25/147) exhibited high-frequency amplification of 19q12, which overlaps with cyclin E1 (CCNE1). We identified that CCNE1 copy number is also highly correlated with gene expression in HuEx (rho = 0.53, 3828112) and U133 (rho = 0.548, 213523_at) datasets. Other regions of amplification also detected in previous studies include 1p31.1 (NEGR1), 8q24.3 and 12q14.1 (CDK4).

We also detected amplification in less commonly reported or novel regions such as 20p13 and 13q34. Amplification of 20p13 has been reported sporadically in previous osteosarcoma array CGH studies, but is not normally recognized as a commonly amplified region in osteosarcoma [28,29]. In our dataset, we identified roughly 20.4% (30/147) of samples with an amplification of 20p13. GISTICs reported region limits in 20p13 overlap with signal-regulatory protein delta (SIRPD) and beta 1 (SIRPB1), two members of the SIRP family of genes. While not much is known about SIRPD, the amplification of SIRPB1 has been associated with primary myelofibrosis, a disease of the bone marrow [30]. Recurrent amplification of 13q34 is a common feature in many types of cancers, but it has not yet been widely explored in osteosarcoma [31]. Several genes located in 13q34 have been associated with a worse prognosis in other cancers, including CUL4A and TFDP1, IRS2 and CDC16 [32,33]. Using gene-level copy number, we identified that CUL4A and TFDP1 copy numbers strongly correlate with expression, indicating a potential mechanism for the change in expression. We did not, however, identify an association with poor outcome, as seen in previous reports.

In addition to the most frequently amplified regions, we also detected 5p15.33 amplification in 40.1% (58/147) and high amplification in 15% (22/147) of cases. This region overlaps with telomerase transcriptase (TERT), an enzyme involved in chromosome end maintenance through the addition of TTAGGG sequences to the 3′ end of telomeres. Chromosome end maintenance, through either telomerase or alternative lengthening of telomeres (ALT), is a pathway through which cancer cells can overcome cell senescence. Unlike other cancers, osteosarcoma has a higher prevalence of ALT in relation to TERT [34]. However, we did not identify a significant correlation between TERT copy number and expression, indicating that TERT expression may be regulated by more complicated mechanisms.

Among the top 10 most recurrent regions with copy number loss, 13q14.2 (RB1) and 3q13.31 (LSAMP/LSAMP-AS3) have also been seen in previous studies. The association between mutations in the Retinoblastoma 1 (RB1) gene and predisposition to osteosarcoma are well known [35,36]. We identified that focal copy number loss of RB1 was also associated with lower expression in both HuEx (rho = 0.52, 3489020) and U133 (rho = 0.53, 203132_at). While we did not identify a correlation between LSAMP/LSAMP-AS3 copy number and expression, deletion of 3p13.31 has been associated with prognosis in multiple osteosarcoma studies and is significantly associated with event-free survival independently (*p*-value = 0.026) in our data [37,38,39]. Three recurrently deleted regions from GISTIC were mapped to 17p13.1, one of which has a wide-peak region 20 Kb away from the beginning of TP53. GISTICs’ wide peak regions encompass a stringent narrow region and the whole region of deletion extends far past its boundaries. To evaluate TP53 deletion on a gene level, we looked at the gene-centric data from GISTIC and found that it was deleted 55.8% of the time (82/147 cases). While TP53 copy number was not directly correlated with gene expression in our data, the mechanism for loss of TP53 function may be achieved through other mechanisms such as mutations or structural abnormalities. Other regions with deletion corresponding with previous studies include 6p27, 6p12, 10q26.3 and 10p15.3.

The novel regions of deletion that we detected include 4q35.2 (DUX4), 16q24.3 (PRDM7, GAS8) and 20q13.33 (MYT1). DUX4 is part of the double homeobox gene family and the deletion identified in our data overlaps with other members of that family. A recent study of small cell osteosarcoma identified 10/36 with a gene fusion involving DUX4 and CIC [40]. Myelin transcription factor 1 (MYT1) is not reportedly involved in normal cell cycle regulation; however, cells with depleted MYT1 have increased CDK1 activity, leading to faster recovery of cell cycle checkpoints due to DNA damage [41]. Therefore, MYT1 depletion in osteosarcoma may contribute to the increased cell proliferation in the presence of chromosomal aberrations.

### 2.4. IGF1R Amplification Is Associated with Higher Expression and Worse Prognosis

To evaluate the clinical significance of these copy number changes, we calculated weighted copy number estimates of GISTICs significantly amplified and deleted regions from segmentation data and evaluated their association with survival (Table 4). Weighted copy number estimates were calculated by overlapping copy number segments with GISTIC regions and weighting the copy number of each segment based on its overlap with the GISTIC region. After Benjamini–Hochberg multiple testing correction, the amplification of both chromosome 8q24.21 (MYC) and 15q26.3 (IGF1R) was associated with poor overall and event-free survival. Initial metastasis is a known prognostic marker for prognosis in many cancers, including osteosarcoma. Accounting for this factor in the survival model rendered 8q24.21 as no longer significant, likely due to MYC’s well-known role in cell growth and association with metastasis. The amplification of 15q26.3, which contains the insulin-like growth factor 1 receptor (IGF1R), is still significant in the presence of initial metastasis for both overall and event-free survival. IGF1R gene copy number was calculated using segmentation data and copy number as a continuous variable is significantly associated with both event-free survival (*p*-value = 4.78 × 10^−5^) and overall survival (*p*-value = 0.0015) with consideration of initial metastasis.

To evaluate the biological effect of IGF1R amplification, we integrated copy number and gene expression data using samples with HuEx array data. We found a strong correlation between copy number and gene expression of IGF1R (Figure 3a). Noting the potential bias of samples with extremely high copy number amplification, we also observed good correlation just within the more numerous moderately amplified samples (CN < 10) (Figure 3b). Given the good association between these two platforms, we further evaluated the clinical significance of IGF1R expression. To achieve this, the probe mapping to IGF1R in the HuEx data was utilized to stratify the samples into high- and low-expression groups, based on a 75th percentile cutoff as a representation of amplified cases. The high expression group showed significantly (Wald test *p* < 0.001) poorer event-free and overall survival, even after accounting for initial metastasis in the model (Figure 3c,d). High IGF1R probe (243358_at) expression in the U133 array cohort modeled as a continuous variable significantly predicted poorer event-free survival (Wald test *p* = 0.016) and overall survival (Wald test *p* = 0.027), but the 75th percentile cutoff did not show significance.

## 3. Discussion

Large-scale genomic characterization studies have improved the overall understanding of many cancers as evidenced by The Cancer Genome Atlas and the International Cancer Genome Consortium projects. While many genomics studies of osteosarcoma have been carried out previously, this study includes the largest number of pediatric osteosarcoma cases to date. As such, the results from this study have benefited from the larger degree of statistical power which accompanies that. We have been able to identify chromosomal aberrations such as amplification of 20p13 and perturbation of 4q35.2, which have been identified sparsely in much smaller osteosarcoma sample sets previously with more significance. In addition, the matched copy number and expression data have allowed us to identify potential mechanisms behind gene expression changes, providing the opportunity for further therapeutic applications.

Copy number aberrations are characteristic of osteosarcoma and many known affected regions were validated in our study, reaffirming the complexity that defines the osteosarcoma genome. We confirmed the association seen in previous studies between copy number and expression in genes such as TOP3A, FLI1, CCNE1, MYC, IGF1R and RB1. While our results demonstrate that most of these recurrent copy number gains/losses show no significant association with survival, we did identify that the amplification of regions in 8q24.21, which overlaps with MYC, and 15q26.3, which overlaps with IGF1R, is associated with gene expression and patient survival. MYC is a well-known transcription factor with roles in cell proliferation, apoptosis and differentiation. MYC amplification and overexpression have been mechanistically linked in many cancers and have been associated with poor prognosis or increased cell proliferation in osteosarcoma cell lines and human tumors [42,43,44,45,46]. We observed that after the inclusion of initial metastasis in the survival model, MYC amplification is no longer prognostically significant. Given the integral role of MYC in cancer metastasis in osteosarcoma, this result is expected.

Our results show that both IGF1R copy number and expression can be used to predict both metastases and survival rates, independent of initial metastatic status. This is consistent with previous reports that implicate increased activity of IGF1R signaling in high-grade osteosarcoma, and that treatment with IGF1R inhibitors can successfully reduce proliferation of osteosarcoma cell lines [47]. IGF1R is a tyrosine kinase membrane receptor that binds both insulin and IGF-2, resulting in insulin receptor substrate protein phosphorylation [48]. The insulin-like growth factor signaling family includes ligands IGF1/IGF2 and receptors IGF1R/IGF2R and activates downstream PI3-kinase and Ras/Raf/ERK signaling pathways [49,50,51]. Osteosarcoma early peak incidence occurs around early adolescence, which coincides with peak expression of circulating IGF1/IGF2 [52]. IGF signaling has been implicated in tumorigenesis and progression of multiple sarcomas including osteosarcoma [53,54]. Interestingly, a previous study tested the ability of IGF1R copy number and expression levels to predict the response to targeted IGF1R antibody therapy in murine xenografts, which found no clear predictive power for either [55]. However, that study lacked the associated clinical outcome data needed to determine whether IGF1R expression or copy number could predict patient survival. IGF1R copy number was also previously shown to be elevated in a small study of 10 primary samples, and this copy number change was correlated with higher IGF1R mRNA expression [7]. Treatment with an anti-IGF1R antibody in mouse xenografts of osteosarcoma showed promising results [55], and this preclinical evaluation provided motivation for a phase II clinical trial testing an anti-IGF1R antibody in combination with mTOR inhibitor in several sarcomas [56]. This trial showed disappointing results. However, upon examination, the trial included only 11 osteosarcoma cases, of which only two cases showed strong immunohistochemistry expression of IGF1R, and no evaluation of IGF1R copy number was used to guide treatment. In combination with these previous findings, our results suggest that IGF1R expression is amplified in a subset of osteosarcoma due to increased IGF1R copy number, and that both can successfully stratify patients into high- and low-risk groups independent of initial metastatic status. Additionally, a Phase 1/2 clinical trial (NCT03746431) targeting IGF1R-overexpressing solid tumors with a radioimmunotherapeutic agent ([225Ac]-FPI-1434) is ongoing, which could prove to be promising for osteosarcoma treatment. Further studies should explore whether levels of IGF1R can be used as a biomarker to guide targeted therapy.

Utilizing mRNA expression data to predict outcomes independent of copy number results, our results indicate that FGFR2 and FIBIN can significantly predict overall survival in both a discovery and validation set. FGFR2 mutations are associated with skeletal abnormalities [57]. Although a previous study implicated allelic loss in FGFR2 [58], allelic loss of this gene in our data was not predictive of either expression or survival (data not shown). FGFR2 expression was found to be reduced in osteosarcoma mouse models [59], but the mechanism by which FGFR2 is downregulated remains elusive. Reduced FGFR2 expression has been observed in multiple cancer types, including bladder [60], prostate [61], and liver [62] cancer, and deleterious mutations in FGFR2 have been implicated in melanoma [63]. In contrast, increased FGFR2 activity is found in several other cancers, including activating mutations in endometrial [64,65], lung [66], and gastric cancer [64], and amplifications and overexpression of FGFR2 in gastric and breast cancer [64]. FGFR2 is a tyrosine kinase receptor that is highly expressed in the cartilage of the growth plate, and plays a vital role in the differentiation of osteoblasts [67]. Furthermore, FGF signaling in the bone has been shown to induce apoptosis in differentiating osteoblasts [68]. Given that FGFR2 downregulation is significantly associated with both poor overall survival and event-free survival in our data, FGFR2 disruption may serve to inhibit differentiation in osteosarcoma. Further studies into how dysregulation of this gene’s expression affects osteosarcoma progression are warranted. FIBIN, a gene essential for pectoral fin bud growth in zebrafish [69], has a poorly understood function in humans, although expression is present in several tissues and may have functions in embryonic development [70]. Given that this gene was found to be differentially expressed in primary/metastatic osteosarcoma cell line pairs [71,72,73], and that it can be used for survival prediction in the present study, further examination of FIBIN in osteosarcoma may provide insights into its function.

The ability to clearly distinguish the large-scale genomic rearrangements and other sequence abnormalities that are characteristic of pediatric osteosarcoma in array-based studies is inferior to the resolution achieved with whole genome and exome sequencing. In addition, other mechanisms such as DNA methylation or microRNA may be used by tumor samples to alter gene expression. Therefore, the need for a comprehensive genomic and epigenomic characterization of osteosarcoma on large sample set is evident and necessary to identifying the underlying pathology and biology of the disease.

Using concurrent genomic platforms profiling on a substantial number of osteosarcoma samples, we were able to establish a correlation between IGF1R copy number and expression and an association with survival. In addition, we were able to identify and validate the prognostic significance of FGFR2 and FIBIN expression in predicting osteosarcoma outcome. Future studies may identify the exact mechanism behind the change in FGFR2 expression, as well as the role of the relatively uncharacterized gene FIBIN.

## 4. Materials and Methods

### 4.1. Patient Samples

Fresh frozen pre-treatment osteosarcoma samples were obtained from the Children’s Oncology Group (COG), the Children’s Cancer Group (CCG) and the Biopathology Center (BPC, Columbus, OH, USA). Before quality control, 226 unique patients were included in this study. Clinical survival data were available for 86% of the cases (*n* = 196) (Appendix A). Tissue was obtained from patients treated on several COG osteosarcoma protocols (P9851, P9754, INT0133, AOST0331, AOST0121) and one CCG protocol (CCG7943). Additional osteosarcoma samples not part of a protocol were obtained from the BPC. Samples eligible for protocols included a combination of initially non-metastatic and metastatic patients. All protocols were approved by the corresponding institutional review board at each participating institution. All patients/parental guardians provided consent to participate in the protocol. All patients involved in this study were under 32 years of age.

### 4.2. Single Nucleotide Polymorphism Array Profiling

DNA genotyping was performed using the Genome-Wide Human Single Nucleotide Polymorphism (SNP) 6.0 array (Affymetrix, Inc., Santa Clara, CA, USA). Arrays were profiled according to manufacturer’s recommendations. Quality control (QC) and genotyping of the SNP 6.0 array was performed in Affymetrix Power Tools (APT, v1.17.0). Affymetrix’s recommended contrast QC cutoff of <0.4 was used to filter out samples with poor quality. Three samples which did not pass this cutoff were excluded from further analysis. Genotyping using Birdseed v2 and allele-specific signal extraction was performed. PennCNV was used to generate log R ratio and b-allele frequencies for each sample, which were input to Allele-Specific Copy Number Analysis of Tumors, v2.1 (ASCAT) to calculate copy number [74,75]. ASCAT accounts for tumor ploidy and aberrant cell fraction in its copy number calculation and, more importantly to this study, can do so in the absence of matched germline DNA. In our study ASCAT successfully resolved 147 samples (Table 5, Appendix A), resulting in 1 additional sample’s exclusion from the study, and subsequent segmentation was performed using ASPCF (allele-specific piecewise constant fitting). Recurrent copy number aberrations were detected using Genomic Identification of Significant Targets in Cancer (GISTIC, v2.0.22) using default settings [76]. Copy number of significant regions from GISTIC were classified into moderate and high levels of amplification and deletion. The distinction of high amplification or deletion was based on GISTIC’s algorithm while moderate amplification and deletion are called using the default input parameter of >0.1.

### 4.3. mRNA Expression Profiling

Affymetrix U133 Plus 2.0 and Human Exon 1.0 ST arrays were used to profile mRNA expression. To assess sample quality, median absolute deviation (MAD) was plotted out for each array and outliers were observed visually. Seven samples were removed from the U133 Plus 2.0 dataset and 1 sample was removed from the Human Exon 1.0 ST dataset (Appendix A). After removal of poor-quality cases, APT was used to perform robust multi-array average (RMA) normalization and calculate gene-centric average expression using annotations provided in NetAffyx (hg19) [77]. Only core probesets with annotated genes were used in downstream analysis. Samples profiled on the Human Exon 1.0 ST array (*n* = 103) were used for the discovery set while samples profiled on the U133 Plus 2.0 array (*n* = 64) were used for validation (Table 5). Patients with data on both arrays were excluded from discovery set. For the discovery sample set, only the top 2.5% most variant probesets by interquartile range (437 probesets, IQR ≥ 1.433) were included in the survival analysis.

### 4.4. mRNA Clustering

In total, 103 samples profiled on the HuEx array were assigned to expression groups by non-negative matrix factorization (NMF) (method = brunet, nrun = 1000, seed = 123456, k = 2 to 6) of the top 10% most variant (1747 probesets, IQR ≥ 1.026) gene-specific probesets using the NMF package [78].

### 4.5. Differential Expression and Pathway Analysis

Differential expression of mRNAs was determined by extracting the features which contributed to NMF clustering. Genes mapping to differentially expressed probes were utilized for pathway enrichment analysis using MetaCore Process Networks enrichment (GeneGo Inc., St. Joseph, MI, USA), with the gene list for all probesets used as background.

### 4.6. Statistical Analysis

Cox proportional hazard regression models [79] adjusted for initial metastasis were used to calculate hazard ratios (HR) for event-free (EFS) and overall survival (OS). Wald test *p*-values were used to determine significance. Survival curves were generated using the Kaplan–Meier estimator [80].

Correction for multiple testing was performed using the Benjamini–Hochberg false discovery rate correction method [81]. Pearson correlation was used to compare copy number and expression platforms. All statistical analysis was performing in R using the stats and survival [82] packages.

## Figures and Tables

**Figure 1 ijms-23-08036-f001:**
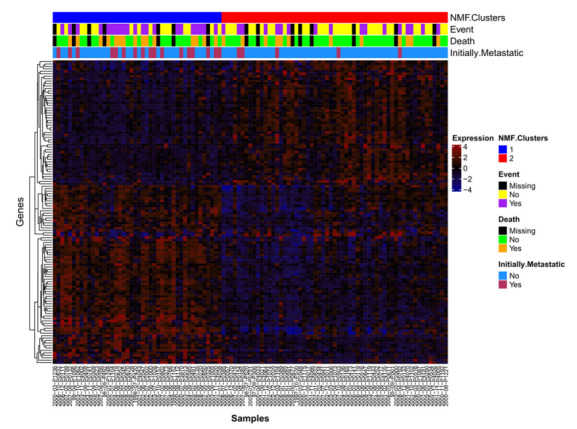
Differentially expressed genes. Samples with HuEx data (*n* = 103) are listed along the x-axis and expression probesets are hierarchically clustered (complete linkage, average distance) on the y-axis. mRNA expression clusters 1 (blue) and 2 (red) are annotated.

**Figure 2 ijms-23-08036-f002:**
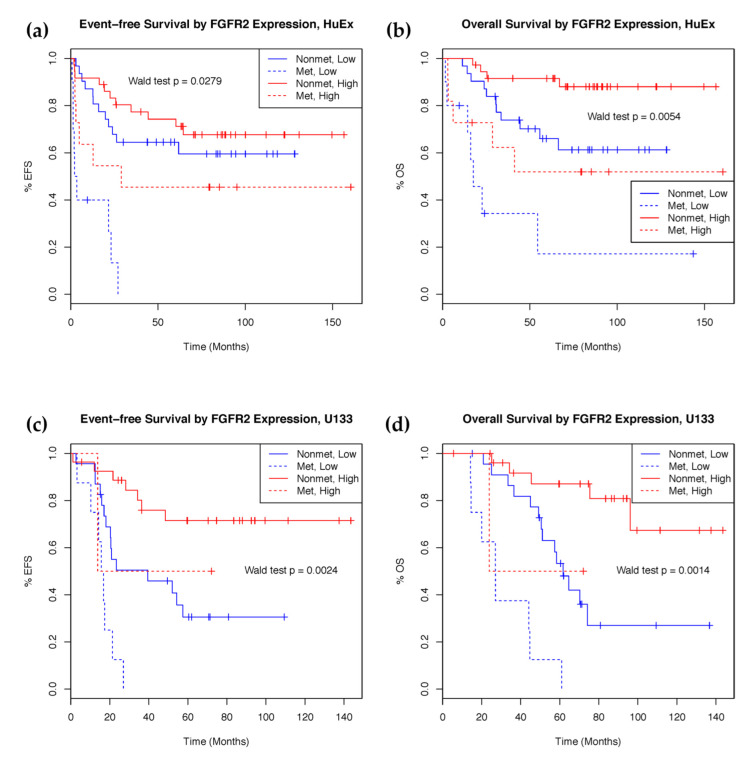
Kaplan–Meier curves showing survival rate difference between patients with high and low expression of FGFR2. (**a**) Event-free survival and (**b**) overall survival in the HuEx discovery cohort (*n* = 88). (**c**) Event-free survival and (**d**) overall survival in the U133 validation cohort (*n* = 60). FGFR2 expression stratified into high (red) and low (blue) expression groups at the 50th percentile.

**Figure 3 ijms-23-08036-f003:**
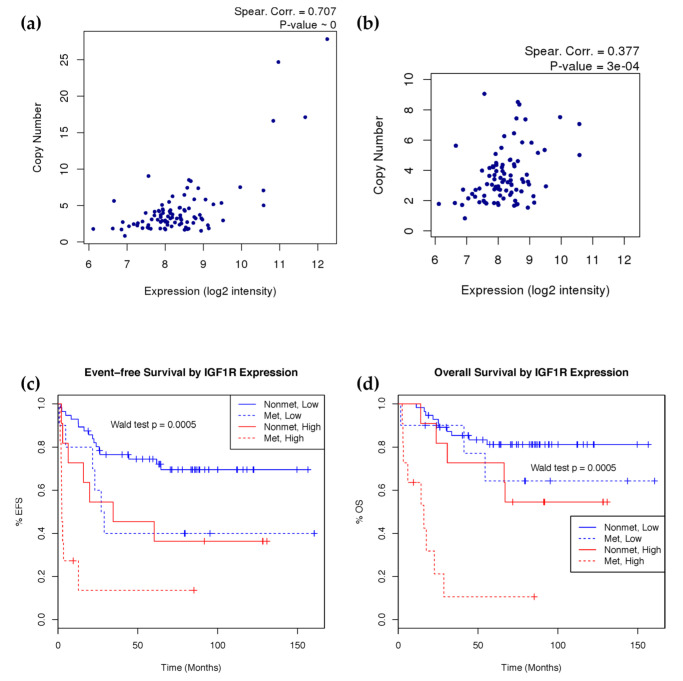
IGF1R copy number and expression correlation and survival analysis. (**a**) IGF1R expression and copy number correlation by spearman correlation using all samples (*n* = 93) and (**b**) after removing high copy number outliers. (**c**) Event-free and (**d**) overall Kaplan–Meier survival curves showing survival rate difference between patients with high and low mRNA expression of IGF1R controlling for metastasis at diagnosis (*n* = 88). Event-free survival models time to first relapse or death. Overall survival models time till death. IGF1R expression stratified into high (red) and low (blue) expression groups at the 75th percentile. Patient’s metastatic status at diagnosis is indicated in solid (non-metastatic) and dotted (metastatic) lines. Wald test *p*-value calculated using the stratified groups in a Cox proportional hazards model including metastasis at diagnosis as a covariate.

**Table 1 ijms-23-08036-t001:** Process networks enriched for differential expression from Metacore.

#	Process Networks	In Data	Total	*p*-Value	FDR	Network Objects from Active Data
1	Cell adhesion_Integrin priming	9	110	1.05 × 10^−5^	6.71 × 10^−4^	ACTA2, G-protein alpha-i family, PIB4, SDF-1, SOS, Actin, Collagen III, PLC-beta, SOS1
2	Cell adhesion_Leucocyte chemotaxis	12	205	1.09 × 10^−5^	6.71 × 10^−4^	G-protein alpha-i family, VCAM1, PIB4, Galpha(i)-specific EDG GPCRs, CCL2, CCL13, SDF-1, CXCL13, Actin, LPA3 receptor, PLC-beta, Galpha(q)-specific EDG GPCRs
3	Development_Regulation of angiogenesis	11	223	1.25 × 10^−4^	4.66 × 10^−3^	FAP48, G-protein alpha-i1, Angiopoietin 1, Ephrin-A receptors, G-protein alpha-i family, IL-6, CCL2, PGAR, N-cadherin, SOS, PLC-beta
4	Development_Blood vessel morphogenesis	11	228	1.52 × 10^−4^	4.66 × 10^−3^	G-protein alpha-i1, Angiopoietin 1, G-protein alpha-i family, VCAM1, ErbB4, Galpha(i)-specific EDG GPCRs, PGAR, SDF-1, PLGF, SOS, HGF receptor (Met)
5	Development_Ossification and bone remodeling	8	157	8.88 × 10^−4^	2.18 × 10^−2^	AEBP1, Frizzled, SFRP4, OSF-2, DMP1, MEPE, Osteomodulin, Bone sialoprotein
6	Cell adhesion_Cadherins	8	180	2.13 × 10^−3^	3.53 × 10^−2^	Frizzled, SFRP4, DKK1, N-cadherin, PTPR-zeta, WIF1, Actin, HGF receptor (Met)
7	Development_EMT_Regulation of epithelial-to-mesenchymal transition	9	225	2.33 × 10^−3^	3.53 × 10^−2^	HGF, ACTA2, Frizzled, G-protein alpha-i family, N-cadherin, SOS, Actin, HGF receptor (Met), Collagen III
8	Development_Skeletal muscle development	7	144	2.46 × 10^−3^	3.53 × 10^−2^	ACTA2, ER81, Actin muscle, ITGA11, ACTG2, Actin, HGF receptor (Met)
9	Inflammation_Protein C signaling	6	108	2.59 × 10^−3^	3.53 × 10^−2^	G-protein alpha-i family, PIB4, Galpha(i)-specific EDG GPCRs, IL-6, Actin, PLC-beta
10	Inflammation_Histamine signaling	8	213	5.95 × 10^−3^	6.28 × 10^−2^	Kappa chain (Ig light chain), G-protein alpha-i family, VCAM1, PIB4, IL-6, CCL2, Actin, PLC-beta

**Table 2 ijms-23-08036-t002:** Survival analysis of significant genes in expression data.

**HuEx Discovery Set (*n* = 88)**									
Probeset ID	Associated Gene(s)	Event-free Survival Model				Overall Survival Model			
		Hazard ratio ^†^	*p*-value ^†,‡^	Corrected *p*-value *	Full-model *p*-value	Hazard ratio ^†^	*p*-value ^†,‡^	Corrected *p*-value *	Full-model *p*-value
3310041	FGFR2	0.718	1.377 × 10^−2^	1.316 × 10^−1^	5.840 × 10^−5^	0.561	5.440 × 10^−4^	3.878 × 10^−2^	8.294 × 10^−6^
3324447	FIBIN	0.698	6.198 × 10^−3^	1.042 × 10^−1^	2.256 × 10^−5^	0.559	5.170 × 10^−4^	3.878 × 10^−2^	6.542 × 10^−6^
3074857	PTN///DGKI	0.729	3.402 × 10^−3^	8.746 × 10^−2^	1.018 × 10^−5^	0.618	6.211 × 10^−4^	3.878 × 10^−2^	3.849 × 10^−6^
3074857	PTN///DGKI	0.729	3.402 × 10^−3^	8.746 × 10^−2^	1.018 × 10^−5^	0.618	6.211 × 10^−4^	3.878 × 10^−2^	3.849 × 10^−6^
**U133 Validation Set (*n* = 60)**									
Probeset ID	Associated Gene	Event-free Survival Model				Overall Survival Model			
		Hazard ratio ^†^	*p*-value ^†,‡^		Full model *p*-value	Hazard ratio ^†^	*p*-value ^†,‡^		Full model *p*-value
211399_at	FGFR2	0.026	1.825 × 10^−3^		2.749 × 10^−5^	0.042	5.347 × 10^−3^		4.658 × 10^−6^
231001_at	FIBIN	0.602	1.129 × 10^−2^		1.216 × 10^−4^	0.595	2.149 × 10^−2^		1.262 × 10^−5^
208408_at	PTN	8.914	4.325 × 10^−2^		3.864 × 10^−4^	24.039	1.163 × 10^−2^		1.985 × 10^−5^
206806_at	DGKI	0.755	3.288 × 10^−1^		1.197 × 10^−3^	0.502	4.080 × 10^−2^		1.741 × 10^−5^

* Multiple testing corrected *p*-values calculated using the Benjamini–Hochberg method. ^†^ Wald-test *p*-values used. ^‡^ All models include initial metastasis at diagnosis as a covariate, reported values based off values for mRNA component.

**Table 3 ijms-23-08036-t003:** Top 10 frequently amplified (red) and deleted (blue) regions from GISTIC (*n* = 147).

Cytoband	Location(Mbs)	Width(Mbs)	Residual q Value	Frequency	High Frequency ‘	Key Genes	
17p11.2	chr17:18.123–18.237	0.114	0	44.9	30.6	TOP3A, FLI1	*
8q24.21	chr8:128.357–128.772	0.415	0	46.9	27.2	MYC	*
20p13	chr20:1.52–1.529	0.009	0	49	20.4		
15q26.3	chr15:99.366–99.408	0.043	0	42.2	20.4	IGF1R	*
1q21.3	chr1:149.996–151.21	1.214	0.001	48.3	19		*
13q34	chr13:105.817–114.882	9.065	0.116	44.2	19		
19p13.2	chr19:12.686–13.498	0.812	0	43.5	18.4		*
6p21.1	chr6:43.323–44.511	1.187	0	40.1	17.7		*
19q12	chr19:30.082–30.306	0.224	0	40.1	17	CCNE1	*
8p11.1	chr8:41.441–50.441	9	0.033	50.3	16.3		
17p13.1	chr17:7.305–7.329	0.024	0.001	58.5			
19q12	chr19:28.283–30.098	1.814	0	58.5			
13q14.2	chr13:48.834–49.065	0.231	0	56.5		RB1	*
17p13.1	chr17:10.372–10.532	0.16	0	56.5			
3q13.31	chr3:116.162–118.625	2.463	0	55.1		LSAMP1, LSAMP-AS1	*
8q24.3	chr8:146.066–146.28	0.214	0	55.1			
17p13.1	chr17:7.611–7.763	0.152	0	55.1			
4q35.2	chr4:190.883–191.154	0.271	0	53.7			
16q24.3	chr16:89.995–90.355	0.36	0	53.1			
20q13.33	chr20:62.735–62.89	0.155	0	53.1			

‘ There were no high-frequency homozygous deletions deleted with significance. * Regions seen in previous studies. High amplification/deletion frequency is determined algorithmically by GISTIC.

**Table 4 ijms-23-08036-t004:** Survival Analysis of Significantly Amplified and Deleted Regions (*n* = 126).

Cytoband	Change	Genes	Full Model				Including Initial Metastasis			
			OS		EFS		OS		EFS	
			*HR*	*p-Value*	*HR*	*p-Value*	*HR*	*p-Value*	*HR*	*p-Value*
15q26.3	Amp	IGF1R	1.110	9.00 × 10^−3^	1.108	3.40 × 10^−2^	1.116	1.30 × 10^−2^	1.093	2.25 × 10^−1^
8q24.21	Amp	MYC, POU5F1B, LOC727677	1.170	1.10 × 10^−2^	1.211	4.00 × 10^−3^				

*OS*—Overall Survival, *EFS*—Event-free Survival, *HR*—Hazard Ratio

**Table 5 ijms-23-08036-t005:** Clinical study of summary set. Only samples used in analyses are displayed.

		**All Samples**		**HuEx Samples**		**U133 Samples**		**Copy Number Only Samples**	
		**#**	**% **	**#**	**% **	**#**	**% **	**#**	**% **
Total		214	100	103	100	64	100	47	100
Gender	Male	122	57	55	53	36	56	31	66
	Female	92	43	48	47	28	44	16	34
Age at Diagnosis	<12	151	71	70	68	46	72	35	74
	>12	63	29	33	32	18	28	12	26
Location	Leg/Foot	183	86	90	87	58	91	35	74
	Arm/Hand	17	8	10	10	2	3	5	11
	Other	11	5	1	1	4	6	6	13
	No Data	3	1	2	2	0	0	1	2
SNP Data	Yes	147	69	93	90	7	11	47	100
	No	67	31	10	10	57	89	0	0
Event	Occurred	87	41	38	37	31	48	18	38
	Censored	100	47	50	49	29	45	21	45
	No Data	27	13	15	15	4	6	8	17
Death	Occurred	68	32	27	26	28	44	13	28
	Censored	119	56	61	59	32	50	26	55
	No Data	27	13	15	15	4	6	8	17
Metastasis at Diagnosis	No	170	79	81	79	54	84	35	74
	Yes	44	21	22	21	10	16	12	26
		**Mean**	**SD**	**Mean**	**SD**	**Mean**	**SD**	**Mean**	**SD**
Age at Diagnosis	Years	13.89	3.78	13.36	3.6	14.4	3.65	14.37	4.25
Follow-up of Survivors	Years	6.47	2.77	6.83	2.78	6.26	2.9	5.87	2.53

## Data Availability

The data presented in this study are available upon request via the COG’s High Dimensional Data (HDD) platform (datarequest@childrensoncologygroup.org).

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
