# Peer review of "Integrated DNA Copy Number and Expression Profiling Identifies IGF1R as a Prognostic Biomarker in Pediatric Osteosarcoma"

_ijms, 2022, doi:10.3390/ijms23148036_

Round 1

Reviewer 1 Report

This study is highly relevant for using a large number of samples to investigate molecular biology of osteosarcoma. The amplification of IGFR1 found in this study may in part explain  why this cancer occurs so often during periors of rapid bone growth. IGFR1 overexpression can also now be targeted using an antibody (FPI-1434) with an alpha emitter, Ac-225, (NCT03746431). Instead of just saying overexpression is associated with worse prognosis, Suggest also mentioning this clincal trial in the discussion.

Reviewer 2 Report

see attached file

Round 2

Reviewer 2 Report

The authors have adequately responded to the reveiwers' comments and I believe the manuscript has improved sufficiently to warrant publication in its present form.